# Combined effect of cell geometry and polarity domains determines the orientation of unequal division

**Benoit G Godard[1,2], Remi Dumollard[1], Carl-Philipp Heisenberg[2]\*, Alex McDougall[1]\***

[1]Laboratoire de Biologie du Développement de Villefranche-sur-mer, Institut de la Mer de Villefranche-sur-mer, Sorbonne Université, CNRS, Villefranche sur Mer, France; [2]Institute of Science and Technology Austria, Klosterneuburg, Austria

**Abstract** Cell division orientation is thought to result from a competition between cell geometry and polarity domains controlling the position of the mitotic spindle during mitosis. Depending on the level of cell shape anisotropy or the strength of the polarity domain, one dominates the other and determines the orientation of the spindle. Whether and how such competition is also at work to determine unequal cell division (UCD), producing daughter cells of different size, remains unclear. Here, we show that cell geometry and polarity domains cooperate, rather than compete, in positioning the cleavage plane during UCDs in early ascidian embryos. We found that the UCDs and their orientation at the ascidian third cleavage rely on the spindle tilting in an anisotropic cell shape, and cortical polarity domains exerting different effects on spindle astral microtubules. By systematically varying mitotic cell shape, we could modulate the effect of attractive and repulsive polarity domains and consequently generate predicted daughter cell size asymmetries and position. We therefore propose that the spindle position during UCD is set by the combined activities of cell geometry and polarity domains, where cell geometry modulates the effect of cortical polarity domain(s).

**\*For correspondence:**
heisenberg@ist.ac.at (C-PhilippH);
alex.mc-dougall@imev-mer.fr (AMcD)

**Competing interest:** The authors declare that no competing interests exist.

## Editor's evaluation

Using the early embryonic divisions of the ascidian *Phallusia mammillata* as a model to investigate mechanisms of unequal cell division, this study convincingly demonstrates that cell shape and cortical domains are cooperating, rather than competing, in order to establish cell size asymmetry, a significant conceptual advance for the field. Their findings provide a new perspective on the roles of cell polarity and shape in the control of spindle positioning, and are of broad interest to cell and developmental biologists.

## Introduction

The century old observation of cells dividing orthogonal to their long axis applies to somatic cells (*Wyatt et al., 2015*) or embryos (*Minc and Piel, 2012*) and is the result of spindle alignment with the longest axis of the cell. This alignment is thought to be mediated by cytoplasmic dynein-dependent pulling forces that scale with astral microtubules length (*Li and Jiang, 2018*; *Minc and Piel, 2012*; *Minc et al., 2011*). Although cell shape often predicts spindle orientation, there are numerous examples where this is not the case in somatic cells (*Finegan and Bergstrahl, 2019*). These deviations from the length-dependent spindle positioning mechanism are usually due to local alteration of microtubule-associated forces by polarity domains. Such polarity domains can

be cortical enrichment of molecular motors such as Dynein (*Grill et al., 2001*; *Kotak and Gönczy, 2013*), polarity proteins such as Ezrin in epithelial cells (*Hebert et al., 2012*; *Korotkevich et al., 2017*), or organelles like yolk granules (*Pierre et al., 2016*) as well as local actomyosin-driven tension (*Scarpa et al., 2018*). This has led to the notion that cell geometry and polarity domains are in direct competition to orient the spindle (*Niwayama et al., 2019*; *Pierre et al., 2016*). Such competition is thought to occur during early embryonic cleavages to shape whole embryos (*Pierre et al., 2016*) and to ensure robust cellular patterning in the mouse blastocyst (*Niwayama et al., 2019*). Yet, whether and how cell geometry and polarity domains compete with each other not only to determine the orientation but also the centering of the mitotic spindle leading to equal or unequal cell divisions (UCDs) remains unclear.

UCD divides the mother cell into two daughter cells of different sizes. This process shapes early embryogenesis in several animal phyla (*Hasley et al., 2017*; *Martín-Durán et al., 2016*; *McDougall et al., 2019*) and sculpts entire organs (*Winkley et al., 2019*). On top of these morphological outcomes, UCDs are often associated with asymmetric segregation of determinants leading to asymmetric cell division creating sibling cells with different cell fates (*Gönczy, 2008*; *Sardet et al., 2005*). While UCDs are a widespread phenomenon in metazoan embryogenesis, the molecular and cellular mechanisms underlying UCD are still being debated.

One major mechanism to generate UCD relies on spindle off-centering, which can be achieved by local cortical pulling on microtubules or microtubules pushing against the cortex. Cortical anchoring of the microtubule minus-end directed motor Dynein, by interaction with LGN/Pins/GPR1-2 and/or NuMA/Mud/Lin-5 (*di Pietro et al., 2016*), exerts pulling forces able to effectively displace centrosomes and mitotic spindles (*Grill and Hyman, 2005*). Such local microtubule pulling forces from the posterior pole of *C. elegans* zygotes leads to UCD during the first cleavage (*Grill et al., 2001*; *Kotak and Gönczy, 2013*; *Redemann et al., 2010*). A similar mechanism is thought to operate also during micromere formation in echinoderm embryos (*Poon et al., 2019*). Cortical pushing can arise when the tip of growing astral microtubule touches the cell cortex creating pushing forces on the spindle (*Garzon-Coral et al., 2016*; *Pavin et al., 2012*). Asymmetric microtubule cortical pushing caused by asters size asymmetry is thought to be implicated in the UCD of the first cleavage in some spiralian embryos (*Ren and Weisblat, 2006*; *Shimizu et al., 1998*) and in ascidian germline precursors (*Costache et al., 2017*). In addition to unequal microtubule pulling and pushing forces, UCD can also emerge from mother cells displaying an anisotropic shape. For instance, during ascidian notochord development, mother cells with conical shape divide along their center of mass, giving rise to daughter cells with unequal size (*Winkley et al., 2019*). Finally, a rare but extensively studied mechanism of UCD is found in neuroblasts of *Drosophila* and *C. elegans* where instead of the spindle the cleavage furrow is off-centered by polarized cortical contractility (*Cabernard et al., 2010*; *Kaltschmidt et al., 2000*; *Ou et al., 2010*).

The early embryo of the ascidian *Phallusia mammillata* has been an emergent model to study mechanisms of UCD due to its invariant cleavage pattern with UCDs in the germline precursors (*Costache et al., 2017*; *Prodon et al., 2010*). Whereas spindle positioning and cleavage pattern in somatic lineages are predominantly determined by cell shape, a macromolecular cortical structure, called the centrosome-attracting-body (CAB), overrides the influence of cell shape to off-center the spindle, thereby inducing three successive UCDs from the 8-cell stage (*Dumollard et al., 2017*). The CAB is inherited by the smaller daughter cells at the posterior pole of the embryo from 16-cell stage onwards, consistent with a decisive role of the CAB in off-centering of the mitotic spindle. However, during the division from the 4- to 8-cell stage, when the CAB is already functional in orienting the mitotic spindle (*Negishi et al., 2007*), the posterior vegetal blastomeres inheriting the CAB are larger than their posterior animal daughters (*Tassy et al., 2006*). Why the CAB at this stage is unable to off-center the mitotic spindle, as found in later divisions, remains unclear.

Here, we show that the UCDs at the third cleavage of ascidian embryo are determined by the combined activities of cell shape and cortical polarity domains. We found that UCDs at the 4-cell stage rely on mitotic spindle tilting in an anisotropic cell shape, which modulates the influence of polarity domains within the dividing cells. We also identified a yet unknown polarity domain localized in the vegetal cortex, devoid of cortical microtubule pulling forces, which is necessary for UCD of anterior cells.

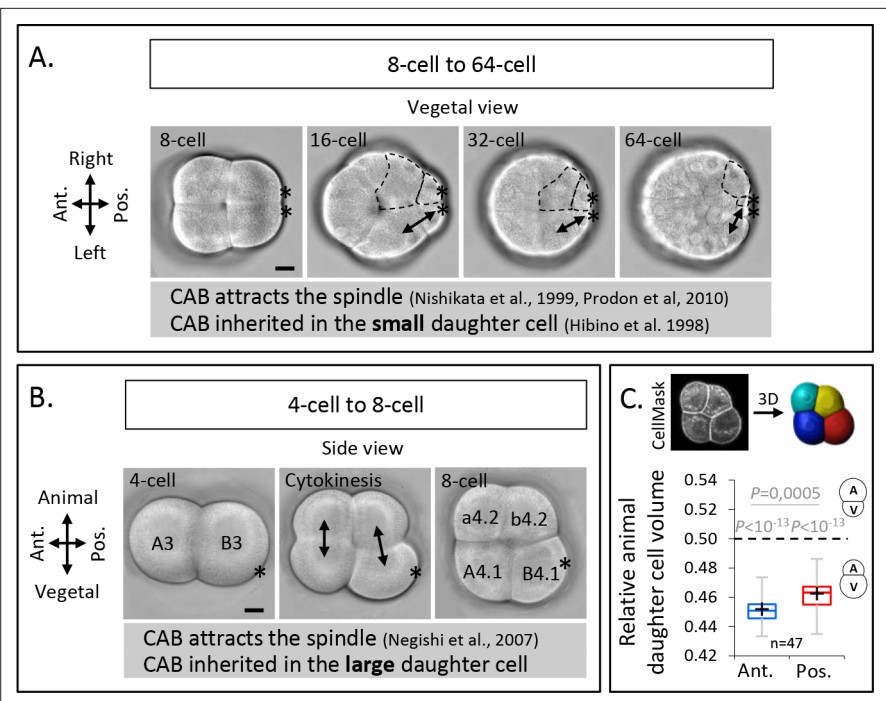

**Figure 1.** Unequal cell division at the third cleavage of ascidian embryos creates larger vegetal blastomeres which inherit the centrosome-attracting-body (CAB). (**A**) Brightfield images of vegetal view of an embryo from 8- to 64-cell stage. The double black arrows on the left side of the embryo mark sister cells and the dashed lines on the right side of the embryo delineate the sister cells of different sizes. Asterisks mark the CAB. (**B**) Brightfield images of left-side view of an embryo positioned in a microwell with animal side on top and posterior to the right at 4-cell stage (left), during cytokinesis (middle) and at 8-cell stage (right). The double black arrows show the sister cells. Asterisks mark the CAB. (**C**) Images (top panel) of left-side view of an 8-cell stage embryo imaged on confocal microscope with CellMask staining and the corresponding 3D reconstruction. Plot (bottom panel) of the ratio of the animal daughter cell volume relative to total daughter cells volume for anterior (in blue) and posterior (in red) lineages at 8-cell stage (center line, median; box limits, upper and lower quartiles; whiskers, min and max; cross, mean). The p values correspond to two-tailed paired Wilcoxon tests for comparisons of anterior or posterior lineages to 0.5 (equal division) and for anterior/posterior lineages comparison. Asterisks mark the CAB. All scale bars, 20 µm.

## Results

### The third cleavage in ascidian embryo creates large vegetal blastomeres by spindle tilting away from the vegetal pole at anaphase onset

The invariant cleavage pattern of ascidian embryos is characterized by three rounds of UCDs in two vegetal-posterior blastomeres from the 8-cell stage onwards, leading to the formation of two small germline precursor cells at the 64-cell stage (*Figure 1A*; *Hibino et al., 1998*). These UCDs are due to spindle attraction by the CAB (*Nishikata et al., 1999*; *Prodon et al., 2010*), which is consistently inherited in the smallest blastomeres that will give rise to the germline precursors. Before these three canonical UCDs, the orientation of the third cleavage is thought to be determined by the CAB attracting one mitotic spindle pole thereby conferring the slanted shape of the 8-cell stage embryo with the CAB located in the pair of vegetal-posterior blastomeres B4.1 (*Figure 1B*; *Negishi et al., 2007*). However, the vegetal blastomeres at the 8-cell stage, containing the CAB, are larger than their animal siblings (*Nishida, 2005*; *Sardet et al., 2007*), raising the question why the CAB is inherited by the larger, rather than the smaller daughter cell as found in subsequent UCDs. To address this question, we first confirmed the size asymmetry between the animal and the vegetal blastomeres in *P. mammillata* embryos by measuring cell volumes from 3D reconstructed live 8-cell stage embryos and found that vegetal blastomeres are indeed larger their animal siblings (*Figure 1C*). Our measurement

also showed that the size asymmetry is slightly but significantly more pronounced in the anterior compared to the posterior pairs of blastomeres (*Figure 1C*).

To determine how the UCDs are set during the 4-cell stage, we used 2D confocal imaging to capture spindle dynamics. Sagittal views (*Figure 2A*) showed that, at metaphase, the spindle in the posterior blastomere B3 is not parallel to the animal-vegetal (AV) axis of the embryo but is slightly tilted by 9.6° as observed previously (*Negishi et al., 2007*). This tilting became more pronounced at anaphase onset with an angle of 21.3° due to the rapid displacement of the vegetal spindle pole toward the CAB (*Figure 2A, B*), suggesting that the CAB attracts the spindle. Consistent with such a role of the CAB at anaphase onset, the spindle was centered along its axis at metaphase and then became vegetally off-centered at anaphase onset (*Figure 2C*). Interestingly, however, cleavage plane prediction on 2D confocal sections showed that the larger size of the vegetal daughter blastomeres (inheriting the CAB) is set at anaphase onset (*Figure 2D*), even though the spindles were attracted toward the CAB (*Figure 2B, C*). This suggests that a mechanism other than spindle off-centering generates the UCD in the posterior blastomere B3 pair. Contrary to the posterior blastomere B3 pair, no clear spindle displacement was observed during the division of the anterior blastomere A3 pair in sagittal views (*Figure 2—figure supplement 1A-C*). We thus monitored spindle position by transversal confocal section to determine whether we could visualize all spindle tilting. Imaging anterior A3 blastomeres in transversal views revealed the presence of a spindle tilting by 5.67° occurring at anaphase onset apparent by a lateral displacement of the vegetal spindle pole away from the vegetal pole (VP; *Figure 2E, F*). In addition, the spindle was slightly off-centered toward the animal pole in anterior blastomeres at anaphase onset (*Figure 2G*). Cleavage plane prediction on 2D confocal sections showed that the larger size of the vegetal daughter blastomere is set at anaphase onset (*Figure 2H*), coinciding with the timing of both spindle off-centering and tilting. Collectively, these findings reveal that spindle tilting at anaphase onset is a common feature in both anterior A3 and posterior B3 blastomeres, preceding UCD.

Since spindle tilting correlates with UCDs at the third cleavage, we performed a virtual analysis to estimate the contribution of spindle tilting to the generation of differently sized daughter cells. To do so, we used transversal confocal sections of anterior A3 blastomeres at anaphase onset from which we extracted the cell contours and the spindle tilting angles measured in *Figure 3E*. We then predicted the relative daughter cell size asymmetry for two virtual cases (*Figure 3A*): in the first case, we first repositioned the spindle parallel to the AV axis and centered it on this axis. In the second case, we applied the measured tilted angles while keeping the virtual spindle centered along its axis (*Figure 3A*). For both cases, we measured the predicted cleavage plane and found that UCD occurred only when tilting is applied to the spindle (*Figure 3A*). This supports the notion that spindle tilting can contribute to UCD independently of spindle off-centering.

To understand better how cell shape affects UCD caused by spindle tilting, we turned to virtual simulations of cleavage plane positioning using two different cell geometries (*Figure 3B*): a circle representing an isotropic cell shape and half-circle representing an anisotropic cell shape. The virtual spindle was then aligned and centered at the center of mass, which led to equal divisions for both shapes. However, when we gradually applied a tilting of the virtual spindle, UCD occurred in the anisotropic geometry as the cleavage plane was gradually shifted from the cente of mass, whereas no UCD occurred in the isotropic geometry as the cleavage plane always crossed the center of mass (*Figure 3B*). This suggests that anisotropy of cell shape can lead to off-centering of the cleavage plane, an effect associated to the activity of spindle tilting in UCD.

## The VP contains a polarity domain which lacks cortical microtubule pulling forces and is necessary to generate large vegetal blastomeres

Next, we sought to understand the mechanisms underlying spindle tilting, apparent by the displacement of the vegetal spindle pole away from the VP in both anterior A3 and posterior B3 blastomeres. To this end, we monitored membrane invaginations after cell cortex weakening as an indicator of microtubule cortical pulling sites, as previously shown in *C. elegans* zygotes (*Redemann et al., 2010*). Membrane-labeled embryos at 4-cell stage treated from Nuclear Envelop Breakdown with low concentration of Cytochalasin B to weaken the actomyosin cell cortex, displayed largely normal embryo shape, but also showed distinct and localized membrane invaginations at the animal pole of both anterior A3 and posterior B3 blastomeres at anaphase onset (*Figure 4A*). Membrane invaginations in

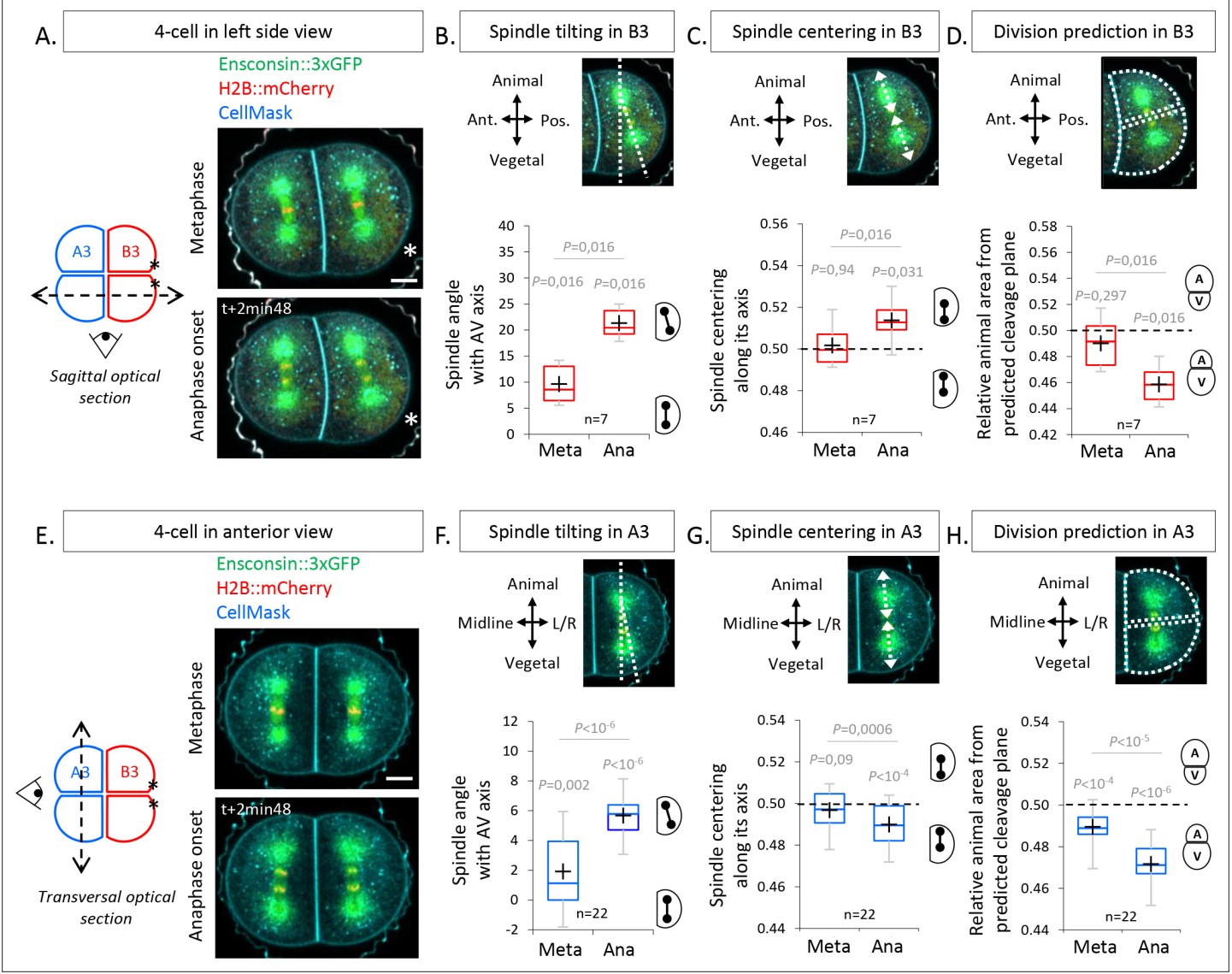

**Figure 2.** The spindles tilt at anaphase onset during unequal cell divisions (UCDs). (**A**) Sagittal views showing spindle position at metaphase and anaphase onset at the 4-cell stage. Schematic illustration (left panel) of a 4-cell stage embryo with the imaging plane in a sagittal view (shown by a dotted double arrow) with anterior blastomeres A3 in blue and posterior blastomeres B3 in red. Fluorescence images of a 4-cell stage embryo in sagittal view (anterior to the left and animal on top) in metaphase (top) and anaphase onset (bottom) previously injected with mRNAs coding for Ensconsin::3xGFP (2 μg/μl, green) and H2B::mCherry (2 μg/μl, red) while plasma membrane is stained by CellMask deep red (1 μg/μl, cyan). The asterisks indicate the position of the centrosome-attracting-body (CAB) in B3 blastomeres. (**B**) Plot of the spindle angle with the animal-vegetal (AV) axis in posterior blastomere B3 measured on sagittal views in metaphase and anaphase. Positive angle means that the mitotic spindles point toward the posterior pole/side of the embryo. (**C**) Plot of the spindle centering along its axis (see Materials and methods for details) measured in posterior blastomere B3 from sagittal views in metaphase and anaphase. 0.5 means centered spindle and above 0.5 means spindle off-centered toward the vegetal pole. (**D**) Plot of the relative animal area after bisection according to the spindle position in posterior blastomere B3 in sagittal view. 0.5 means animal and vegetal area are equal and below means vegetal area is larger. (**E**) Transversal views showing spindle position in anterior blastomeres A3 in metaphase and anaphase onset at the 4-cell stage. Schematic illustration (left panel) of 4-cell stage embryo showing the transverse optical plane across the anterior blastomeres with anterior blastomeres A3 in blue and posterior blastomeres B3 in red. Fluorescence images of 4-cell stage embryo in transversal view (animal on top) in metaphase (top) and anaphase onset (bottom) previously injected with Ensconsin::3xGFP (2 μg/μl, green) and H2B::mCherry (2 μg/μl, red) mRNAs while plasma membrane is stained with CellMask deep red (1 μg/μl, cyan). (**F**) Plot of the spindle angle with the AV axis in anterior blastomere A3 in transversal view in metaphase and anaphase. Positive angle means that the mitotic spindles point away from the midline. (**G**) Plot of the spindle centering along its axis in anterior blastomere A3 in transversal view in metaphase and anaphase. 0.5 means a centered spindle and below 0.5 indicates a spindle off-centered toward the animal pole. (**H**) Plot of the relative animal area after bisection according to the spindle position in anterior blastomere A3 in transversal view. 0.5 means animal and vegetal area are equal and below means vegetal area is larger. All the p values correspond to two-tailed paired Wilcoxon tests for comparisons to 0.5 (equal division) or for metaphase/anaphase comparisons. The box

*Figure 2 continued on next page*

*Figure 2 continued*

plots are built with center line, median; box limits, upper and lower quartiles; whiskers, min and max; cross, mean.

The online version of this article includes the following figure supplement(s) for figure 2:

**Figure supplement 1.** Spindle dynamics in anterior blastomeres in sagittal view.

Cytochalasin B-treated embryos were dependent on microtubules since they were completely abolished when embryos were treated with Nocodazole to depolymerize microtubules (*Figure 4—figure supplement 1*). Furthermore, while no membrane invaginations were detected at the VP, some invaginations were observed at the CAB site (visible by a dense membrane staining) in posterior B3 blastomeres and at the anterior/lateral side in anterior A3 blastomeres (*Figure 4A*). These findings are consistent with the previously reported ability of the CAB to effectively pull on the spindle at 4-cell

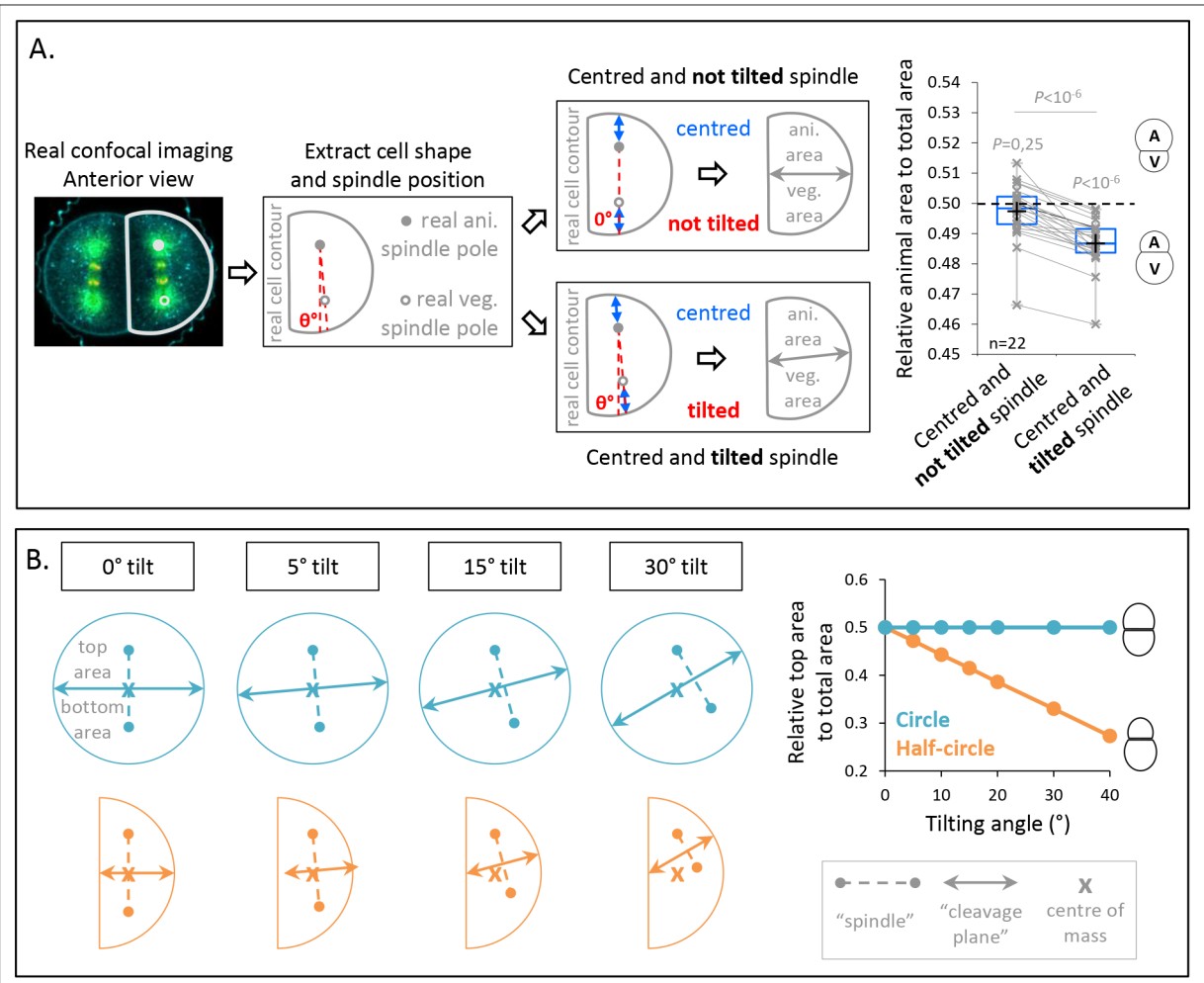

**Figure 3.** Spindle tilting in an anisotropic cell geometry induces unequal cell division (UCD) by displacing the cleavage plane from the cell center of mass. (**A**) Simulation of the effect of a tilted but centered spindle on the bisection of the cell. Schematic illustration of the simulations realized using real cell contour of anterior blastomeres from transversal view. First scenario, the spindle is positioned parallel to the animal-vegetal (AV) axis (not tilted) and centered along this axis (top panel). Second scenario, the spindle tilting is applied (tilted) while kept centered along its axis (bottom panel). Plot of the relative animal area to total cell area after bisection according to the spindle angle using the tilting angles measured at anaphase onset (*n* = 22) (right panel) (center line, median; box limits, upper and lower quartiles; whiskers, min and max; cross, mean). 0.5 means animal and vegetal area are equal and below means vegetal area is larger. Gray lines connect simulation for the same real cell contour and tilting angle. The p values correspond to two-tailed paired Wilcoxon tests for comparisons to 0.5 (equal division) and for comparison between the two conditions. (**B**) Theoretical evaluation of spindle tilting in different cell geometries. Schematic illustration of the division plane from a gradually tilted but centered spindle (the top spindle pole remains fixed) in an isotropic shape (circle) (top panel) and in a anisotropic shape (half-circular shape) (bottom panel). Plot of the relative top area to total area after bisection according to the spindle tilting (left panel). 0.5 means top and bottom area are equal and below means bottom area is larger.

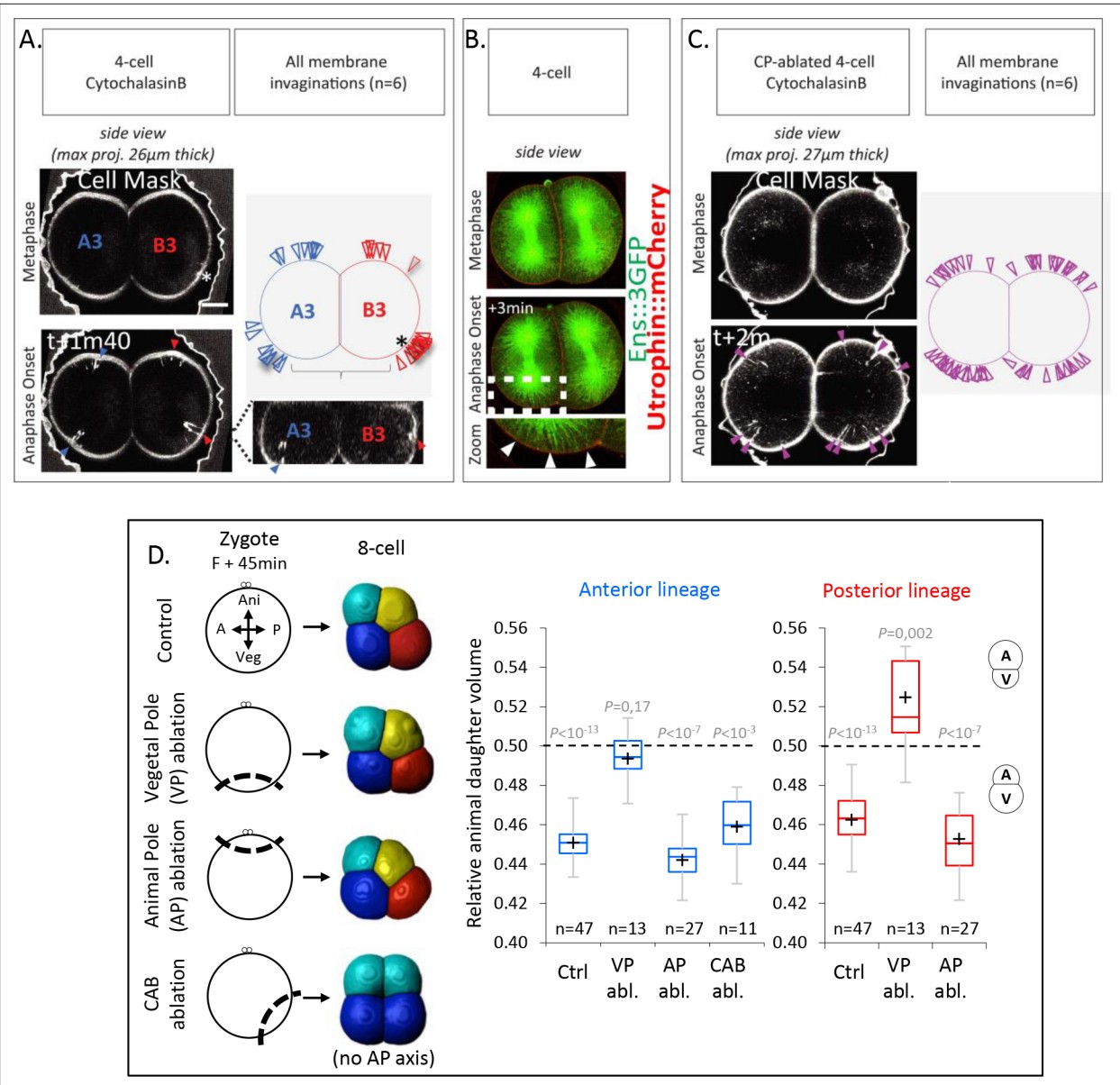

**Figure 4.** The vegetal cortex is a polarity domain lacking microtubule cortical pulling forces and is necessary for making larger vegetal blastomeres. (**A**) Cortical pulling sites on astral microtubules at mitosis during the 4-cell stage observed as plasma membrane invaginations in embryos with a weakened cortex. Maximum projection (26 µm thick) of confocal images of 4-cell stage embryo in sagittal view (animal on top) during mitosis (left panel). Embryos were treated with cytochalasin (15 µM) from Nuclear Envelop Break Down to soften the cell cortex during mitosis and the plasma membrane is stained with CellMask deep red (at 1 µg/µl). Orthogonal section (bottom right) at the level indicated by the dashed lines showing the location of first invaginations at the site of centrosome-attracting-body (CAB) in B3 and in lateral position in A3. Schematic illustration summarizing all the membrane invaginations observed in six embryos (top right). The blue and red arrow heads indicate sites of membrane invaginations for the anterior blastomere A3 and the posterior blastomere B3, respectively. Asterisks show the CAB localization. Scale bar, 20 µm. (**B**) Astral microtubules reach the cortex at anaphase onset. Fluorescent confocal images of 4-cell stage embryo in sagittal view (anterior to the left and animal on top) in metaphase (top) and anaphase onset (middle) injected with Ensconsin::3xGFP (2 µg/µl, green) and Utrophin::mCherry (2 µg/µl, red) mRNAs. Enlarged view (bottom) corresponding to the area delineated by the dashed box where white arrow heads indicate astral microtubules contacting the cortex at the vegetal pole at anaphase onset. Scale bar, 20 µm. (**C**) Vegetal pole (VP)-ablated early zygotes (F + 15 min) were treated with Cytochalasin B (15 µM) from Nuclear Envelop Break Down to soften the cell cortex during mitosis, and the plasma membrane is stained with CellMask deep red (at 1 µg/µl). Invaginations at the 4-cell stage in a pair of blastomeres are displayed together with a schematic illustration summarizing all invaginations observed (n = 6). (**D**) Ablations of cell cortex at zygote stage and resulting blastomeres size at 8-cell stage. Schematic illustration (left panel) of the ablation performed at zygote stage after the second phase of ooplasmic segregation (animal on top) and the corresponding 8-cell stage visualized in 3D reconstruction in sagittal view (anterior to the left and animal on top). Plots of the ratio of animal daughter cell volume relative to total daughter cell volume for anterior lineage (in

*Figure 4 continued on next page*

*Figure 4 continued*

blue, middle) and posterior lineages (in red, right) at 8-cell stage measured from 3D reconstructions in control embryos and embryos with the VP (*F* + 45 min), or the animal pole (AP) ablated, or CAB ablated (center line, median; box limits, upper and lower quartiles; whiskers, min and max; cross, mean). The p values correspond to two-tailed paired Wilcoxon tests for comparisons of anterior or posterior lineages to 0.5 (equal division). Asterisks show the CAB localization.

The online version of this article includes the following figure supplement(s) for figure 4:

**Figure supplement 1.** Membrane invaginations are abolished by depolymerizing microtubules.

**Figure supplement 2.** Yolk distribution in oocyte and 4-cell stage; gastrulation phenotypes after zygote microsurgeries.

stage (*Negishi et al., 2007*), but also indicate that the VP of both anterior A3 and posterior B3 blastomeres is devoid of microtubule cortical pulling forces even though the spindle was pointing toward this domain. Importantly, the observed cortical pulling forces are sufficient to explain the spindle tilting motion observed in those blastomeres, suggesting that no other mechanism may be involved.

The absence of cortical pulling at the VP could be due to the absence of microtubules contacting the cortex at this site. An AV yolk gradient with higher yolk granule density at the VP of blastomeres has previously been proposed to account for the UCDs at the third cleavage of ascidian embryos by limiting microtubule growth toward the VP (*Pierre et al., 2016*). However, staining of yolk granules showed a homogeneous distribution in oocytes and no such AV gradient at the 4-cell stage, with yolk evenly accumulating at the cell periphery (*Figure 4—figure supplement 2A and B*). Consistent with this, we found that astral microtubules in live embryos, which become detectable in mitosis from anaphase onset, were able to reach the vegetal cortex of these cells (*Figure 4B*). We thus conclude that the lack of microtubule pulling forces at the VP, as evidenced by the lack of membrane invagination in Cytochalasin B-treated embryos, is not due to microtubules being unable to reach the cortex, but rather that this cortex is devoid of microtubule pulling force generators.

Our data thus point to the possibility that the VP contains a polarity domain characterized by a lack of microtubule pulling force generators, leading to spindle tilting and UCD. To further test whether the vegetal cortex may indeed not exert pulling forces, we ablated the VP of early zygotes (Fert + 15 min) to determine if membrane invaginations would be present in VP (*F* + 15 min)-ablated embryos (*Figure 4C*). Consistent with our assumption of the VP not exerting pulling forces, we found that in VP-ablated embryos, membrane invaginations were present at both poles of the embryo (*Figure 4C*). To further test whether the VP indeed contains a polarity domain, we conducted microsurgeries at the late zygote stage, when embryonic polarities are established after two phases of ooplasmic segregations (*Roegiers et al., 1999*), followed by 3D reconstruction at the 8-cell stage for cell volume measurements (*Figure 4D*) to measure the impact of cortex ablation. To validate the ablation of the VP, we verified that the determinant for gastrulation (located in the vegetal cortex of the zygote) was indeed removed by confirming the absence of endoderm invagination at the 128-cell stage, and we also noticed the presence of a blastocoel (*Figure 4—figure supplement 2C*; *Nishida, 1996*). Such VP-ablated embryos had an equal division in the anterior lineages and an inverted UCD in the posterior lineages at the third cleavage (*Figure 4D*). In contrast, proper UCDs leading to larger vegetal blastomeres were maintained after control ablation of the animal pole or the posterior pole (containing the CAB) (*Figure 4D*; note that gastrulation occurred normally in those control ablation, *Figure 4—figure supplement 2C*). These microsurgery experiments suggest that the VP contains a polarity domain necessary for proper spindle tilting and, consequently, the generation of larger vegetal blastomeres.

## Anisotropic cell shape modulates the effect of polarity domains on UCD

Our results so far suggest that cell shape anisotropy during mitosis at the 4-cell stage—in conjunction with spindle tilting—is critical for UCD and determines the interaction between the spindle and the surrounding cortical polarity domains. The anisotropic shape of mitotic blastomeres at the 4-cell stage is characterized by large cell–cell contacts and a pronounced long axis of the cell oriented along the AV axis of the embryo (*Figure 2A, E*). This elongated cell shape is due to these cells not undergoing mitotic rounding when transiting from interphase to mitosis, apparent by cell sphericity decreasing rather than increasing, as expected for cells undergoing mitotic rounding (*Figure 5—figure*

*supplement 1A*). Given such pronounced shape anisotropy of the mitotic blastomeres, we hypothesized that decreasing their shape anisotropy should perturb UCD (*Figure 3B*), leading to a loss of daughter cell size asymmetry in the anterior blastomeres A3, and an inversion of cell size asymmetry in the posterior blastomeres B3, where the spindle is attracted by the CAB. To test this hypothesis, we conducted manual isolation of pairs or single blastomeres at the 4-cell stage to increase the sphericity of those blastomeres when entering mitosis outside of the spatial confinement of the embryo (*Figure 5A, B*). Mitochondrial staining allowed us to identify the corresponding animal and vegetal blastomeres after the cell division of isolated blastomeres as vegetal blastomeres inherit more mitochondria (*Figure 5—figure supplement 1B*; *Chenevert et al., 2013*). Consistent with our expectations, 3D cell reconstructions before and after the division showed that an increase in mother cell sphericity was associated with a decrease of the UCDs in the anterior lineages and an inversion of the size asymmetry in the posterior lineages (*Figure 5A* and *Figure 5—figure supplement 1C-E*). Live observation of the mitotic spindle in isolated anterior A3 and posterior B3 blastomeres further showed that spindle tilting and spindle off-centering still occurred between metaphase and anaphase, suggesting that the activity of the polarity domains is maintained in such isotropic cell shape (*Figure 5C-H*). Yet, contrary to the situation in anisotropic blastomeres, the spindle tilting is incapable of inducing UCD in these isotropic blastomeres, and, consequently, spindle off-centering led to a strongly reduced UCD in anterior A3 blastomeres and an inversion of UCD in posterior B3 blastomeres (*Figure 5B*). Collectively, these findings underline the importance of cell shape anisotropy for UCD at the 4-cell stage.

Given the importance of cell shape anisotropy for blastomere UCD, we asked how the orientation of cell's longest axis influences UCD. Our results so far have revealed the existence of different polarity domains within the blastomeres at the 4-cell stage, with the CAB in posterior cells exerting microtubule pulling forces, and a newly identified domain at the vegetal cortex of both anterior and posterior blastomeres devoid of those pulling forces. We thus hypothesized that the cell geometry, influencing spindle orientation, represents a critical parameter modulating the interaction of the spindle with the surrounding cortex and its polarity domains. In particular, we hypothesize that blastomere geometry determines the general orientation of the spindle along the AV axis of the embryo, and that the presence of a vegetal polarity domain lacking microtubule pulling forces causes vegetal spindle pole displacement from the VP, spindle tilting and, consequently, off-centering of the cleavage plane and an UCD generating a larger vegetal cell (*Figure 6A*). Furthermore, in the posterior blastomeres, spindle tilting would be even more pronounced due to the presence of the CAB pulling on the spindle (*Figure 6A*). In order to test this hypothesis, we sought to physically manipulate the cell geometry to induce a different long axis that would in turn modify the contribution of the two cortical polarity domains (vegetal cortex and CAB). Our model predicted that an AV compression would reduce the influence of the vegetal polarity domain and increase the influence of the CAB due to reduced blastomere AV elongation and thus spindle alignment along this axis (*Figure 6B*). By using CellMask to label endocytosed vesicles accumulating around spindle poles and thus outlining spindle orientation (*Dumollard et al., 2017*), we monitored spindle position in compressed embryos (*Figure 6B*). As expected, AV compression led to spindle alignment in the plane of the compression (perpendicular to the AV axis) and its displacement in direction to the posteriorly located CAB in posterior B3 blastomeres and, consequently, an inversion of the directionality of the UCD in the posterior B3 blastomeres with the CAB now being inherited in the smaller daughter cells (*Figure 6B*). Taken together, these findings suggest that the combined activities of cell geometry and polarity domains determine spindle off-centering and UCD.

## Discussion

Different mechanisms have been proposed to generate UCDs among metazoans such as spindle off-centering (i.e., *C. elegans*, some spiralians) (*Grill and Hyman, 2005*; *Kotak and Gönczy, 2013*; *Pavin et al., 2012*; *Ren and Weisblat, 2006*; *Shimizu et al., 1998*), polarized cortical contractility (*D. melanogaster* and *C. elegans* neuroblast) (*Cabernard et al., 2010*; *Kaltschmidt et al., 2000*; *Ou et al., 2010*), or cell shape anisotropy (i.e., some spiralians, ascidian notochord) (*Toledo-Jacobo et al., 2019*; *Winkley et al., 2019*). Our findings reveal spindle tilting in an anisotropic cell shape as a yet unknown mechanism determining UCD within the early ascidian embryo. Importantly, this mechanism can in principle work independently of spindle off-centering, since it can even override the impact of spindle

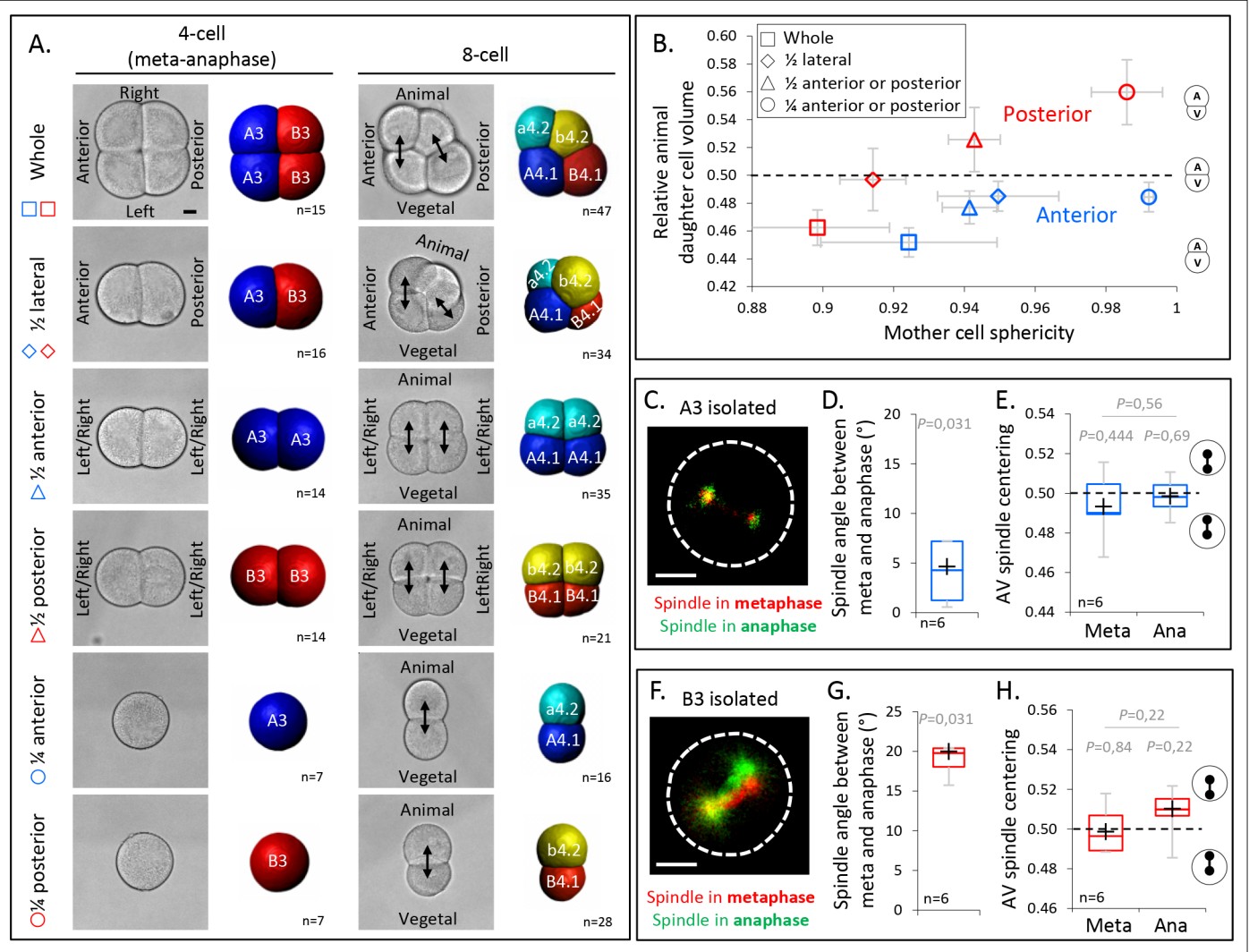

**Figure 5.** Decreasing cell shape anisotropy of the mitotic blastomeres changes the unequal cell divisions (UCDs). (**A**) Left panel: brightfield images and corresponding 3D reconstruction (anterior blastomeres [A3] in blue, posterior blastomeres [B3] in red) of 4-cell stage embryos in mitosis either whole or after isolations of blastomeres (1/4) or pair of blastomeres (1/2). Right panel: brightfield images and corresponding 3D reconstruction at 8-cell stage (anterior blastomeres [a4.2, A4.1] in blue and cyan, posterior blastomeres [b4.2, B4.1] in red and yellow) either whole or after isolation of blastomeres (1/4) or pair of blastomeres (1/2). Isolations were performed at the 4-cell stage. Note that the whole embryo at 4-cell stage is oriented with an animal view whereas all other embryos are shown in side views. Scale bar, 20 µm. (**B**) Plot showing the ratio of animal daughter cell volume relative to total daughter cells volume at 8-cell stage as a function of their mother cell sphericity in mitosis for control embryos and isolated blastomeres (1/4) or pair of blastomeres (1/2) for anterior (blue) and posterior lineages (red). Error bars are standard deviation. (**C**) Fluorescence images of isolated anterior A3 blastomere previously injected with mRNAs coding for Ensconsin::3xGFP (2 µg/µl, green). The spindle is labeled in red and green for the time corresponding to metaphase and anaphase onset, respectively. (**D**) Plot of the angle between the spindle at metaphase and anaphase onset in isolated anterior A3 blastomeres. (**C**) Plot of the spindle centering along its axis (see Materials and methods for details) measured in isolated anterior A3 blastomere in metaphase and anaphase. 0.5 means centered spindle and above 0.5 means spindle off-centered toward the vegetal pole. (**E**) Fluorescence images of isolated posterior B3 blastomere previously injected with mRNAs coding for Ensconsin::3xGFP (2 µg/µl, green). The spindle is labeled in red and green for the time corresponding to metaphase and anaphase onset, respectively. (**F**) Plot of the angle between the spindle at metaphase and anaphase onset in isolated posterior B3 blastomeres. (**C**) Plot of the spindle centering along its axis (see Materials and methods for details) measured in isolated posterior B3 blastomere in metaphase and anaphase. 0.5 means centered spindle and above 0.5 means spindle off-centered toward the vegetal pole/centrosome-attracting-body (CAB). All the p values correspond to two-tailed paired Wilcoxon tests for comparisons to 0.5 (equal division) or for metaphase/anaphase comparisons. The box plots are built with center line, median; box limits, upper and lower quartiles; whiskers, min and max; cross, mean. Scale bar, 20 µm.

The online version of this article includes the following figure supplement(s) for figure 5:

**Figure supplement 1.** Cell sphericity during 4-cell stage; Blastomere identification and morphometric analysis of isolated blastomeres.

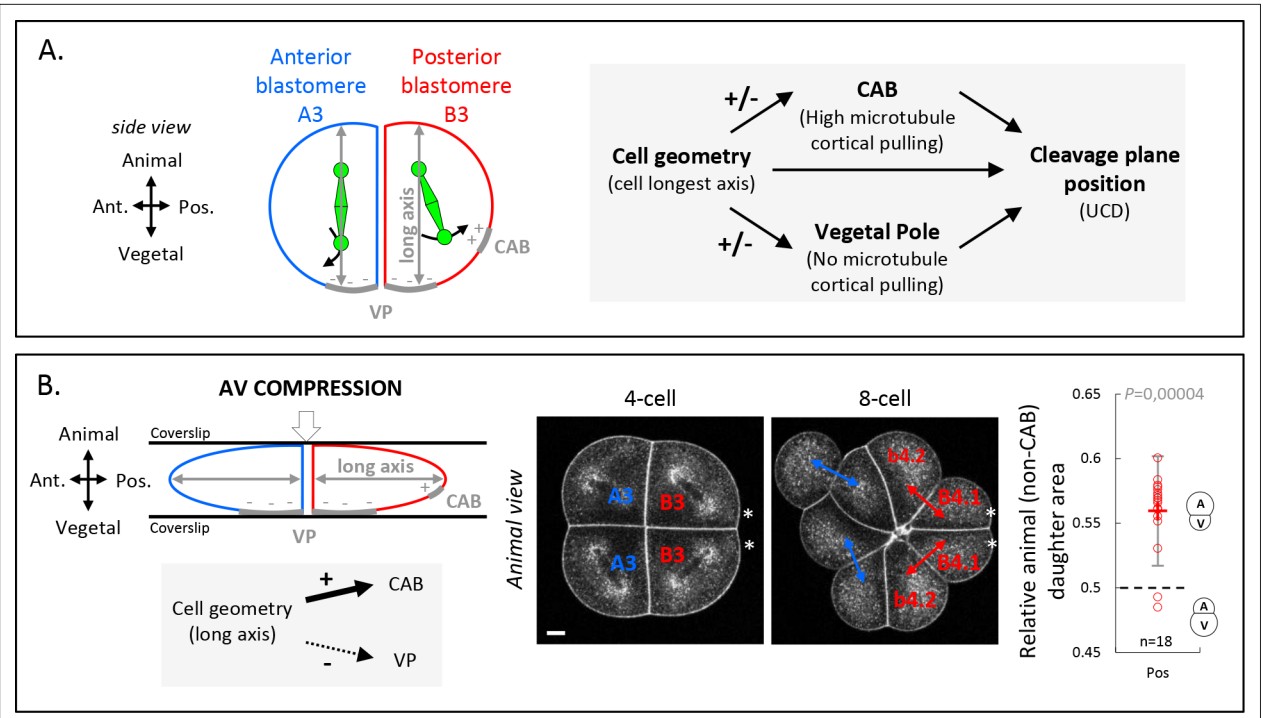

**Figure 6.** Anisotropic cell shape influences the spindle interaction with polarity domains and together determines the orientation of unequal cell division (UCD). (**A**) Schematic illustration (left panel) of a 4-cell stage embryo in sagittal view (anterior to the left, animal on top) showing the spindle interaction with the different cortical polarity domains. Illustration of the cleavage plane positioning mechanism (right panel) where the cleavage plane position is a result of the interplay between cell geometry, the posterior centrosome-attracting-body (CAB) polarity domain, and the vegetal pole (VP) polarity domain. (**B**) Animal-vegetal (AV) compression of 4-cell stage embryo. Schematic illustration a 4-cell stage embryo in sagittal view with an AV compression (top left panel) to bring the long axis of the cell toward the CAB and away from the vegetal pole domain (bottom left panel). Optical confocal section of anteroposterior compressed embryo (middle panel) at 4-cell stage anaphase and at 8-cell stage with membrane stained with CellMask orange (at 1 µg/µl). Plot of the relative animal daughter cell area (right) for posterior lineages at 8-cell stage in AV compressed embryos (center line, median; box limits, upper and lower quartiles; whiskers, min and max; cross, mean). The p values correspond to two-tailed paired Wilcoxon tests for comparisons to 0.5 (equal division). Asterisks show the CAB localization. Scale bar, 20 µm. $n = 18$.

pulling by the CAB in the 4-cell stage ascidian embryo, but relies on cells not undergoing pronounced mitotic rounding. It is thus conceivable that spindle tilting also determines UCD in other cell types that do not undergo complete mitotic rounding as in early *Xenopus* embryos (***Chalmers et al., 2003***), in the enveloping cell layer of zebrafish embryos (***Campinho et al., 2013***) or in *Drosophila* follicle cell epithelium (***Aguilar-Aragon et al., 2020***).

The reasons why cells undergo different degrees of mitotic rounding and, thus, might use spindle tilting as a mechanism for UCD, could be manifold. In ascidian 4-cell stage embryos, cells upon entry into mitosis slightly increase cortical tension at their cell–medium interface (***Godard et al., 2020***), but also retain low cortical tension at their cell–cell interfaces evident by the presence of large cell–cell contacts (***Turlier and Maître, 2015***), leading to a highly anisotropic cell shape during mitosis. This effect is restricted to early cleavage stages, as from the 16-cell stage onwards cells undergo more pronounced mitotic rounding by decreasing their apical cortical tension, which also allows mitotic cells to be deformable by extrinsic forces implementing tension-oriented cell divisions (***McDougall et al., 2019***). Thus, in ascidians spindle tilting as a mechanism determining UCD dependent of cell shape anisotropies might be largely restricted to early cleavage stages, while at later stages other mechanisms, such as spindle off-centering by polarity domains, might become more important, as found in the germline precursors with spindle off-centering by the CAB (***Costache et al., 2017***; ***Prodon et al., 2010***).

A key finding of our study is that anisotropic cell shape functions in concert with different polarity domains modulating microtubule pulling activities during UCD. In ascidian embryos, the CAB is thought to attract one spindle pole toward the posterior cortex of the embryo, thereby generating

three successive rounds of UCD from the 8-cell stage onwards where the CAB is consistently inherited by the smaller daughter cell, eventually giving rise to two germline precursors at the 64-cell (*Negishi et al., 2007*; *Prodon et al., 2010*). However, a major paradox has been that, although the CAB is already active at the 4-cell stage (*Negishi et al., 2007*), it is not inherited in the smaller, but rather the larger daughter cell after UCD at the 8-cell stage. Our data resolve this paradox by showing that (1) in addition to the CAB, there is a vegetal polarity domain with no microtubule pulling forces and (2) cell shape anisotropy at mitosis directs spindle alignment to the vegetal cortex away from the CAB, thereby mitigating the activity of the CAB in pulling the spindle toward the posterior of the cell. Importantly, this not only explains how the CAB can be inherited by the larger cell during the 4- to 8-cell stage cleavage, but also more generally provides insight into the way by which UCD is determined by the combined activities of cell shape and polarity domains.

The forces exerted on microtubules to position the spindle in early ascidian embryo were proposed to originate from cytoplasmic dynein (*Pelletier et al., 2020*) which has also been proposed to constitute the main mechanism operating in very large cells (*Kotak et al., 2013*). However, in smaller cells like in ascidians, the astral microtubules can easily reach the cell cortex, which can then also apply pulling forces on those microtubules. Thus, a combination of cytoplasmic and cortical microtubule forces is likely involved in precise spindle positioning in early ascidian embryos. The nature of the cortical force generator remains to be elucidated. However, the cortical pulling occurring at anaphase onset suggests a possible link to a CDK-1-dependent loss of NuMA phosphorylation known to drive its cortical localization (*Pierre et al., 2016*). In addition to cytoplasmic and cortical pulling provided by microtubules, we also provide evidence for the existence of a yet unknown polarity domain at the VP of the embryo, which is characterized by the lack of pulling activity. Whereas polarity domains are usually considered as cortical sites that provide a local elevated source of microtubule cortical pulling forces, as shown for the CAB, the vegetal polarity domain clearly displays reduced microtubule pulling activity. The molecular basis for the reduced pulling activity of the vegetal cortical polarity domain remains to be determined; previous studies and our own preliminary data suggest that this polarity domain is established after the second phase of ooplasmic segregation in the zygote and is probably neither a subcortical accumulation of yolk (*Pierre et al., 2016*) nor a cortical gradient of the PI3 kinase shown in the ascidian *H. roretzi* (*Takatori et al., 2015*).

The role of cell geometry in cell division orientation is well established (*Howard and Garzon-Coral, 2017*; *Li and Jiang, 2018*; *Minc and Piel, 2012*; *Minc et al., 2011*; *Pavin et al., 2012*) and is now understood as the result of microtubule length-dependent force generation which positions the spindle along the longest axis of the cell (*Pierre et al., 2016*). Likewise, there is now ample evidence for the presence of cortical polarity domains, able to locally influence the forces exerted on microtubules and thus bias the position of the spindle (*Niwayama et al., 2019*; *Pierre et al., 2016*). However, cell geometry and cortical polarity domains have generally been proposed to be in competition, with one of them eventually dominating in positioning of the spindle (*McDougall et al., 2014*). Our study, in contrast, suggests that cell geometry and polarity domains act in concert to determine spindle positioning, with cell geometry modulating the effect of cortical polarity domains by influencing the position of the spindle relative to those polarity domains. It will be interesting to determine whether the activity of cortical polarity domains can be affected by cell shape also in other systems where cells remain nonspherical during mitosis and contain cortical polarity domains. In *C. elegans* embryogenesis, for instance, where different polarity domains were shown to affect UCD, embryo deformation alters the precise placement of the cleavage planes (*Yamamoto and Kimura, 2017*), pointing at the intriguing possibility that the coaction of cell shape and polarity domains represents an evolutionary conserved principle determining UCD.

## Materials and methods
### Animal collection/maintenance and embryo handling

Adults individuals of the European ascidian *P. mammillata* were either purchased from the Roscoff marine station (France) and kept in artificial sea water (ASW) or collected from Sète (France) and kept in filtered natural sea water at 16°C. Eggs and sperm were collected by dissection. Eggs were dechorionated in sea water implemented with Trypsin (T8003, Sigma-Aldrich) at 0.1% for 1–2 hr, and sperm was activated in pH 9.0 sea water prior fertilizations (*Schindelin et al., 2012*).

### Embryo orientation, imaging, and 3D reconstruction

Transparent circular microwells were used to orient embryos and image them with a confocal microscope. For this, a homemade array of little pillars made in PDMS was used to cast circular microwell in a polymer of the same refractive index as water (MY-134, MY Polymers) in a glass bottom Petri dish (P35G-1.5-14C, MatTek). For 3D reconstruction, images of 1.48 $\mu m^3$ voxel size were acquired every 2 min on an inverted Leica TCS SP5, equipped with a HC PL APO CS2 ×40/1.10 water immersion objective (Leica), a Hybrid detector and a homemade cooling stage set at 19°C. The images were processed with Ilastik software to segment the membrane signal followed by an intensity threshold applied with ImageJ (*Yasuo and McDougall, 2018*). 3D reconstruction was performed on the segmented images with Imaris 9.0 (Bitplane), which provided measurement of the cell volume and cell sphericity. The relative animal daughter cell volume corresponds to the volume of the animal cell divided by the total volume of the daughter cells (animal and vegetal).

### Blastomere isolation

Blastomere or pair of blastomeres were isolated at 4-cell stage using a thin glass filament gently inserted in between blastomeres observed on an Olympus SZX16 stereomicroscope. After isolation, the blastomeres were transferred in ASW supplemented with CellMask orange (1 µg/ml; C10045, Invitrogen) and MitoTracker deep red (1 µg/µl; M22426, Invitrogen) and imaged on a confocal microscope as for 3D reconstruction detailed above.

### Microtubules live imaging and spindle dynamics quantification

Dechorionated oocytes were injected with *Ensconsin::3xGFP* and *H2B::mCherry* or *Utrophin::mCherry* polyA-tailed mRNA at 2 µg/µl, synthetized using mMessage mMachine SP6 Transcription Kit (Invitrogen, AM1340) on a microinjection setup mounted on an Olympus SZX16 stereomicroscope as in *Schindelin et al., 2012* except that single embryos were oriented in an agarose circular well made from a homemade PDMS pillar microarray. After a minimum 4 hrs of incubation, injected oocytes were fertilized and early 4-cell stage embryos were placed in a circular microwell (see above) containing ASW and CellMask deep red at 1 µg/µl (1 µg/ml; C10046, Invitrogen). Confocal time-lapse acquisition of embryos left/right side or anterior facing the objective were performed on an inverted Leica TCS SP5, equipped with a HC PL APO CS2 ×40/1.10 water immersion objective (Leica), a Hybrid detector and a homemade cooling stage set at 19°C. All the quantification of spindle position were measured using ImageJ software (*Schindelin et al., 2012*) on 2D confocal section allowing to precisely capture the anaphase onset. The spindle angle is relative with the AV axis, the spindle centering is the ratio of the distance between chromosomal plate and animal cell edge divided by the distance between the chromosomal plate and the vegetal cell edge, measured on the spindle axis. The division prediction is calculated as the ratio of animal and vegetal areas with a cleavage plane positioned perpendicularly to the spindle at its midzone.

### Simulations of the impact of spindle tilting on cleavage plane Positioning

To simulate spindle tilting without spindle off-centering, a virtual repositioning of the spindle was performed using ImageJ (*Schindelin et al., 2012*) on real confocal images of anterior views of 4-cell stage embryos (*Figure 2E*). Using the time point of anaphase onset, the spindle was first aligned with and centered along the AV axis. Then, a tilting was applied by using the measured angles for the corresponding embryo, while still keeping the spindle centered along its axis. For these two scenarios, the cell area was divided perpendicularly from the center of the spindle and the relative animal area was measured. For the full virtual simulation of the effect of spindle tilting in a circular (nonconstrained) and a half-circular (constrained) geometry, all the scenarios were performed using the different drawing tools of ImageJ (*Schindelin et al., 2012*).

### Imaging of microtubules cortical pulling sites

Embryo were placed in ASW with 15 µM of Cytochalasin B (Sigma-Aldrich, C6762), 10 µM Nocodazole (Sigma-Aldrich, M1414), and CellMask deep red at 1 µl/ml (Invitrogen, C37608) at the time of Nuclear Envelop Break Down (NEBD) in circular microwell (see above). Fast time-lapse confocal acquisition of the embryo left or right side were performed on an inverted Leica TCS SP5 equipped with a HC

PL APO CS2 ×40/1.10 water immersion objective (Leica). Around 40–60 s after the first membrane invagination occurred, corresponding to the time of anaphase, a maximum projection of selected planes was done to generate the image on which counting of the sites of membrane invaginations was performed.

### Zygote microsurgeries

Ablations of the cell cortex were performed on zygotes either before or after the second phase of ooplasmic segregation (CAB ablation, VP ablation [Fert + 15 min], and VP ablation [Fert + 45–50 min]) on an Olympus SZX16 stereomicroscope microinjection setup where the injection needle was replaced by a blunt-end glass pipette with an internal diameter of 40 µm and a 30° bent angle (Biomedical Instruments). The VP or the animal pole was aspirated within the pipette and sectioned from the embryo with a thin glass needle. The volume of the ablated cytoplasm was around 12% of the oocyte volume. At 8-cell stage, ablated embryos were placed in ASW with CellMask orange (1 µg/ml; C10045, Invitrogen) in a circular microwell for 3D reconstruction (see above). After 8-cell stage, the embryo were kept on the microscope to monitor the phenotypes associated with each microsurgery at gastrula stage.

### Embryo compression

For anteroposterior (AP) compression, early 4-cell stage embryo were placed in a dish containing ASW with CellMask orange (1 µg/ml; C10045, Invitrogen) with a piece of PDMS sculpted with a cavity of rectangular geometry. The compression was realized by placing a coverslip on top of the PDMS. The embryos placed in a cavity providing the proper AP compression were then imaged on an upright Leica TCS SP5 or SP8 equipped with a HC PL APO ×20/0.8 (Leica). For AV compression, early 4-cell stage embryos were placed in ASW with CellMask orange (1 µg/ml; C10045, Invitrogen) on a glass slide. Then a coverslip was gently added on the top and pressed under a stereoscope until the compression was observed. Embryos were then imaged on an inverted Leica TCS SP5 or SP8 equipped with a HC PL APO ×20/0.8 (Leica).

### Quantification and statistical Analysis

Graphical analysis of data and statistics was performed with Excel (Microsoft). All the p values were calculated with either two-tailed unpaired Mann–Whitney or two-tailed paired Wilcoxon tests. No statistical method was used to predetermine sample size and the experiments were not randomized.

## Acknowledgements

We thank members of the Heisenberg and McDougall groups for technical advice and discussion. We are grateful to the Bioimaging and Nanofabrication facilities of IST Austria and the Imaging Platform (PIM) and animal facility (CRB) of Institut de la Mer de Villefranche (IMEV), which is supported by EMBRC-France, whose French state funds are managed by the ANR within the Investments of the Future program under reference ANR-10-INBS-0, for continuous support. This work was supported by a collaborative grant from the French Government funding agency Agence National de la Recherche to McDougall (ANR 'MorCell': ANR-17-CE 13-0028) and the Austrian Science Fund to Heisenberg (FWF: I 3601-B27).

## Additional information

### Funding

| Funder | Grant reference number | Author |
|---|---|---|
| Agence Nationale de la Recherche | ANR-17-CE 13-0028 | Alex McDougall |
| Austrian Science Fund | FWF: I 3601-B27 | Carl-Philipp Heisenberg Alex McDougall |

| Funder | Grant reference number | Author |
|--------|------------------------|--------|

The funders had no role in study design, data collection, and interpretation, or the decision to submit the work for publication.

## Author contributions

Benoit G Godard, Investigation, Writing - original draft; Remi Dumollard, Investigation, Writing - review and editing; Carl-Philipp Heisenberg, Supervision, Writing - original draft, Writing - review and editing; Alex McDougall, Supervision, Writing - review and editing

## Author ORCIDs

Remi Dumollard http://orcid.org/0000-0002-8444-0630
Carl-Philipp Heisenberg http://orcid.org/0000-0002-0912-4566
Alex McDougall http://orcid.org/0000-0003-0324-3836

## Decision letter and Author response

Decision letter https://doi.org/10.7554/eLife.75639.sa1
Author response https://doi.org/10.7554/eLife.75639.sa2

## Additional files

### Supplementary files
• Transparent reporting form

### Data availability
All main figures are supplied with data used to generate the figures.

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
