## [Editor Report]

Using the early embryonic divisions of the ascidian *Phallusia mammillata* as a model to investigate mechanisms of unequal cell division, this study convincingly demonstrates that cell shape and cortical domains are cooperating, rather than competing, in order to establish cell size asymmetry, a significant conceptual advance for the field. Their findings provide a new perspective on the roles of cell polarity and shape in the control of spindle positioning, and are of broad interest to cell and developmental biologists.

---

## [Decision Letter]

[Editors' note: this paper was reviewed by Review Commons.]

---

## [Author Response]

Reviewer #1 (Evidence, reproducibility and clarity (Required)):Summary:Unequal cell division is the key developmental process by which one cell divides into two daughter cells of different sizes. This sculpts tissues and embryos into their final shape, but also give rise to the great variety of cell types in an organism as it is usually coupled with asymmetric partitioning of cell fate determinants. A central area of investigation in the field is to understand the molecular mechanisms underlying unequal cell division. Indeed, it is well established that cell shape controls the position of the mitotic spindle (cell divides along their long axis), but, at the same time, cortical polarity domains are also known to be able to orient and offset the mitotic spindle. Thus, a key question is to know the relative contribution of these two phenomena during development (does one win over the other? or do they collaborate?).In this elegant study, Godard and colleagues address this question directly in the early ascidian embryo (Phallusia mammillata) at the 4-cell stage. This model system is a nicely suited for the question as at that stage, the embryo is both polarized and has an anisotropic shape. The authors combine quantitative live imaging, microdissection and biophysical perturbations to establish that cell shape and polarity domains cooperate to establish cell-size asymmetry in daughter cells. In particular, they characterise cortical domains of the membrane that behave as spindle attractors or repellent. I particularly liked the spectacular demonstration of the reversal of cell-size asymmetry upon decreasing cell shape anisotropy (Figure 5B). This is such a beautiful experiment!Overall, the study is particularly elegant technically, as to get comprehensive quantitative data, the authors put embryos in microfabricated structures to enable live confocal imaging followed by robust image processing using pixel-classification methods to automatically segment embryos in 3D to measure cell volume data. Similarly, all the results are carefully quantified, use an adequate number of replicates to support the claims and using statistics to test significance when appropriate. Last, all the data are clearly presented and the data carefully interpreted. The authors made a particularly nice job of covering a vast literature in their introduction, as well as to put their data in context in the discussion.Minor comments:My main concern is the establishment of the Vegetal Pole as a spindle "repellent" (Figure 4). While the beautiful vegetal pole ablation experiment clearly demonstrates it does something that goes in the right direction, without the spindle tilting data, it's difficult to rule out other potential explanations. Similarly, the evidence that membrane invaginations in the cytochalasin treated embryos correspond to pulling via microtubules comes from another system (*C. elegans*). While this is probably also true in ascidians, I think it would bring to check that this is indeed the case.I think that adding one extra piece of evidence to strengthen this part would benefit the paper if the authors want to keep this as a strong point (and I think they should).– For instance, the authors could add the spindle data for control versus vegetal pole ablation (i.e. no more tilt for anterior cells, and some residual, CAB-mediated, tilt for posterior).

We published previously that the internalized vesicles accumulate at spindle poles and thus use these visualizations to follow spindle positions. This has been clarified in the text.

Line 294:

“By using Cell Mask to label endocytosed vesicles accumulating around spindle poles and thus outlining spindle orientation (Dumollard et al., 2017), we monitored spindle position in compressed embryos (Figure 6B).”

– Alternatively, the authors could treat embryos ablated for the vegetal pole with cytochalasin and see that now invaginations appear where the vegetal pole should have been.

We performed this experiment. By ablating the vegetal contraction pole (CP) then performing the cytochalasinB experiment we found that invaginations now occurred at both poles – new data added to new Figure 4 part C. Text has been added to Figure 4 Figure Legend.

Line 501:

“(C) Vegetal pole (VP)-ablated early zygotes (F+15min) were treated with Cytochalasin B (15µM) from Nuclear Envelop Break Down to soften the cell cortex during mitosis, and the plasma membrane is stained with Cell Mask Deep Red (at 1µg/µl). Invaginations at the 4-cell stage in a pair of blastomeres is displayed together with a schematic illustration summarizing all invaginations observed (n=6).”

Text added to main body. Line 224:

“To further test whether the vegetal cortex may indeed not exert pulling forces, we ablated the vegetal pole (VP) of early zygotes (Fert + 15 min) to determine if membrane invaginations would be present in VP (F+15 min)-ablated embryos (Figure 4C). Consistent with our assumption of the vegetal pole not exerting pulling forces, we found that in VP-ablated embryos, membrane invaginations were present at both poles of the embryo (Figure 4C).”

– The author could also depolymerise microtubules in cytochalasin-treated embryos (this should remove invaginations). Any of these control experiments (or anything else the author might think of or already have) would help I think (no need to do all three of course).

We performed this experiment. By depolymerizing microtubules the invaginations were completely absent. New supplementary Figure S4 has been added.

Also see text added at Line 197:

“Membrane invaginations in Cytochalasin B-treated embryos were dependent on microtubules since they were completely abolished when embryos were treated with Nocodazole to depolymerize microtubules (Figure S4).”

Also see Figure Legend, Line 619:

“Figure S4 – related to Figure 4: Membrane invaginations are abolished by depolymerizing microtubules

Cortical pulling sites on astral microtubules at mitosis are abolished when microtubules are depolymerized during the 4-cell stage. […] Embryos were treated with Cytochalasin B (15µM) and Nocodazole (10µM) from Nuclear Envelop Break Down to soften the cell cortex during mitosis, and the plasma membrane is stained with CellMask Deep Red (at 1µg/µl). Time is shown. N=8.”

Other (cosmetic) comments:– I really like the thought experiments ("simulation") that the authors use throughout the paper to extrapolate what the cell size asymmetry would have been if the spindle was not tilted given a specific cell shape. Since this is rare, it would help the reader to provide a bit more explanation on how this is done in the main text at the first occurrence (L146), especially because the authors did a drawing to help (top panel in 2D). Something like "when we predicted the expected daughter cell area asymmetry based on the geometry of the mother cell and the tilt/centring of the spindle (Figure 2D, top panel), we found that the larger size of the vegetal cell…"

See additional sentence that has been added Line 152:

“We thus monitored spindle position by transversal confocal section to determine whether we could visualize all spindle tilting.”

– It might be worth insisting on the fact that the absence of tilting seen in Sagittal view for A3 (L150 and Figure S1) is just a matter of differential orientation of the division plane in 3D between B3 and A3. The way I understood it, we know there must be some tilt in some view, as there is a volume asymmetry of the anterior cells (Figure 1C). It's just a matter of finding which 2D view is the most appropriate to measure it.

The text has been re-written to clarify this point Line 150:

“Contrary to the posterior blastomere B3 pair, no clear spindle displacement was observed during the division of the anterior blastomere A3 pair in sagittal views (Figure S1A-S1C). […] Imaging anterior A3 blastomeres in transversal views revealed the presence of a spindle tilting by 5.67° occurring at anaphase onset apparent by a lateral displacement of the vegetal spindle pole away from the vegetal pole (Figure 2E and 2F).”

– Line 156: reference to "Figure 3G", which does not exist. I think the authors mean 2G.

Corrected to 2G.

– Figure 3A and Figure 4C: I think the cartoon showing the cell size asymmetry (right-hand side) is inverted. Shouldn't it be the vegetal cell larger for a relative animal area to total area below 0.5?

Thank you. This has been corrected.

– Figure 3A, right panel. It might be useful to plot the "real" cell (i.e. off-centred and titled) for comparison.

We chose this option because we wanted to re-enforce the notion that throughout the article we compare real embryos and modelling. However, if the reviewer prefers that we change the schematic with a real image we shall do that.

Reviewer #1 (Significance (Required)):I think this study is a highly significant conceptual advance for the field, as it shows that cell shape and cortical domains are cooperating, rather than competing, in order to establish cell size asymmetry. This paper will thus pave the way towards a mechanistic and molecular understanding of the relative contribution of both phenomena in various context and organisms.This paper will be of interest for the developmental biology, cell biology and cytoskeleton/polarity fields.My fields of expertise are developmental cell biology, polarity and cytoskeleton dynamics.Reviewer #2 (Evidence, reproducibility and clarity (Required)):The process of unequal cell division (UCD) has been extensively investigated during development. Here, the authors explore the mechanisms of unequal cell division during the third cleavage plane of the ascidian Phallusia mammillata. Past findings show that from the 16-cell stage onward, the Centrosome-Attracting-Body (CAB) polarizes cell division to override the influence of cell shape leading to off-centered spindle and unequal daughters, the smaller daughter cell inheriting the CAB. Yet, and as previously reported during the third division, the larger blastoderm inherits the CAB, suggesting the evidence of a yet unidentified mechanism controlling UCD. Using a combination of live-imaging, quantitative measurements, as well as mechanical and surgical embryos manipulations, the authors propose that UCD in the third cleavage results from a spindle tilt due to the concerted action of a vegetal unknown polarity factor and of the cell shape anisotropy of the A3 and B3 blastomere.Major comments:1. Apart from the data of the figure 6B, the reported results are highly convincing. In figure 6B, a smaller number of experiments are performed (n=8) and statistical significance is borderline. This mechanical manipulation being central to the authors' conclusion and model, I would therefore suggest increasing the number of experiments to consolidate the author's conclusions.

We have since increased the number of experiments.

See line 564: “n = 18”.

2. The authors utilized qualitative model and prediction. While this is usually satisfactory, applying physical modeling to explore rigorously the contribution of polarity, spindle tilting, and cell shape anisotropy would be important to reinforce that cell shape and polarity are sufficient to reproduce third cleavage UCD. For example, confirming the dependencies and trends observed in Figure 5B and Figure 6B by physical modelling would be highly relevant to strengthen the authors' cooperative model.

Although a physical model would be ideal, we would need to develop and/or extend a model that simultaneously allows 3D modelling integrating cell polarity with two additional cortical domains to recapitulate our experimental findings. Moreover, for such model being useful, it would require experimental access to input parameters, such as the individual strength of different polarity domains, which for technical reasons are difficult to obtain within the timeframe of a normal revision. That said – and as much as we agree with the referee that such model would be desirable – developing and applying such model would go beyond the scope of the present study.

3. The authors used only one approach to demonstrate that the vegetal pole does not generate pulling forces. Could the authors perform laser ablation to further establish the lack of vegetal pulling forces? Could this polarity domain be more resistant to F-actin depolymerization? Is the vegetal pole enriched in F-actin or Myosin?

As an alternative to laser ablation we have now ablated the vegetal cortex of zygotes to remove the vegetal cortical domain. By ablating the vegetal contraction pole then performing the cytochalasin experiment we found that invaginations now occurred at both poles – new data added to new Figure 4 part B.

Text has been added to Figure 4 Figure Legend. Line 501:

“(C) Vegetal pole (VP)-ablated early zygotes (F+15min) were treated with Cytochalasin B (15µM) from Nuclear Envelop Break Down to soften the cell cortex during mitosis, and the plasma membrane is stained with Cell Mask Deep Red (at 1µg/µl). Invaginations at the 4-cell stage in a pair of blastomeres is displayed together with a schematic illustration summarizing all invaginations observed (n=6).”

Text added to main body, Line 224:

“To further test whether the vegetal cortex may indeed not exert pulling forces, we ablated the vegetal pole (VP) of early zygotes (Fert + 15 min) to determine if membrane invaginations would be present in VP (F+15 min)-ablated embryos (Figure 4C). Consistent with our assumption of the vegetal pole not exerting pulling forces, we found that in VP-ablated embryos, membrane invaginations were present at both poles of the embryo (Figure 4C).”

Line 706:

“Ablations of the cell cortex were performed on zygotes either before or after the second phase of ooplasmic segregation (CAB ablation, VP-ablation (Fert +15min) and VP-ablation (Fert +45 to 50 min))”.

We visualized actin distribution in live embryos with Utrophin::mCherry and did not detect any differences in actin at the poles of cells during the 4 to 8-cell stage. Since we did not pursue this line of analysis we did not include these data in the article.

Reviewer #2 (Significance (Required)):My expertise is not in ascidian biology. The cited methods described in the text and Materials and methods indicate that the authors likely used classical techniques and approaches. The authors frame the novelty of their manuscript in the opposition between the mechanism of simple orientation versus UCD: during spindle orientation, cell polarity overrides cell shape. While this has been observed in some context, as referenced by the authors, it is also known that cell shape and polarity are often aligned and may therefore co-promote spindle orientation. The cited literature is mainly limited to embryonic development and should further acknowledge numerous studies on spindle orientation in cell culture or tissues.

We have amended the first paragraph to include literature relating to somatic cells.

Line 43, the start of the paragraph now reads:

“The century old observation of cells dividing orthogonal to their long axis applies to somatic cells (Wyatt et al., 2015) or embryos (Minc and Piel, 2012) and is the result of spindle alignment with the longest axis of the cell. et al.[…] Such polarity domains can be cortical enrichment of molecular motors such as Dynein (Grill et al., 2001; Kotak and Gönczy, 2013), polarity proteins such as Ezrin in epithelial cells (Hebert et al., 2012; Korotkevich et al., 2017), or organelles like yolk granules (Pierre et al., 2016) as well as local actomyosin driven tension (Scarpa et al., 2018).”